# EMPIRICAL ANALYSIS OF REPRESENTATION LEARNING AND EXPLORATION IN NEURAL KERNEL BANDITS

## ABSTRACT

Neural bandits have been shown to provide an efficient solution to practical sequential decision tasks that have nonlinear reward functions. The main contributor to that success is approximate Bayesian inference, which enables neural network (NN) training with uncertainty estimates. However, Bayesian NNs often suffer from a prohibitive computational overhead or operate on a subset of parameters. Alternatively, certain classes of infinite neural networks were shown to directly correspond to Gausian processes (GP) with neural kernels (NK). NK-GPs provide accurate uncertainty estimates and can be trained faster than most Bayesian NNs. We propose to guide common bandit policies with NK distributions and show that NK bandits achieve state-of-the-art performance on nonlinear structured data. Moreover, we propose a framework for measuring independently the ability of a bandit algorithm to learn representations and explore, and use it to analyze the impact of NK distributions w.r.t. those two aspects. We consider policies based on a GP and a Student's t-process (TP). Furthermore, we study practical considerations, such as training frequency and model partitioning. We believe our work will help better understand the impact of utilizing NKs in applied settings.

## 1 INTRODUCTION

Contextual bandit algorithms, like upper confidence bound (UCB) (Auer et al., 2002) or Thompson sampling (TS) (Thompson, 1933), typically utilize Bayesian inference to facilitate both representation learning and uncertainty estimation. Neural networks (NN) are increasingly applied to model non-linear relations between contexts and rewards (Allesiardo et al., 2014; Collier & Llorens, 2018). Due to lack of a one-size-fits-all solution to model uncertainty with neural networks, various NN models result in trade-offs in the bandits framework. Riquelme et al. (2018) compared a comprehensive set of modern Bayesian approximation methods in their benchmark, and observed that state-of-the-art Bayesian NNs require training times prohibitive for practical bandit applications. Classic approaches, on the other hand, lack complexity to accurately guide a nonlinear policy. Further, the authors showed that the neural-linear method provides the best practical performance and strikes the right balance between computational efficiency and Bayesian parameter estimation.

Bandit policies balance exploration and exploitation with two terms: (1) a mean reward estimate and (2) an uncertainty term (Lattimore & Szepesvári, 2020). From a Bayesian perspective those two terms represent the first two moments of a posterior predictive distribution, e.g. a Gaussian process (GP). Research on neural kernels (NK) has recently established a correspondence between deep networks and Gaussian processes (GP) (Lee et al., 2018). The resulting model can be trained more efficiently than most Bayesian NNs. In this work we focus on the conditions in which NK-GPs provide a competitive advantage over other NN approaches in bandit settings. Even though NKs have been shown to lack the full representational power of the corresponding NNs, they outperform finite fully-connected networks in small data regimes (Arora et al., 2019b), and combined with GPs, successfully solve simple reinforcement learning tasks (Goumiri et al., 2020). NK-GPs provide a fully probabilistic treatment of infinite NNs, and therefore result in more accurate predictive distributions than those used in most of the state-of-the-art neural bandit models. We hypothesized that NKs would outperform NNs for datasets possessing certain characteristics, like data complexity, the need for exploration, or reward type. The full list of considered characteristics can be found in Tab. 2. We ran an empirical assessment of NK bandits using the contextual bandit benchmark proposed by

Riquelme et al. (2018). The obtained results show that NKs provide the biggest advantage in bandit problems derived from balanced classification tasks, with high non-linear feature entanglement.

Measuring empirical performance of bandits may require a more detailed analysis than rendered in standard frameworks, in which the task dimensions are usually entangled. Practitioners choosing the key components of bandits (e.g. forms of predictive distributions, policies, or hyperparameters), looking for the best performance on a single metric, may be missing a more nuanced view along the key dimensions of representation learning and exploration. Moreover, interactions between a policy and a predictive distribution may differ depending on a particular view. In order to provide a detailed performance assessment we need to test these two aspects separately. For most real-world datasets, this is not feasible as we do not have access to the data generating distribution. In order to gain insight into the capability to explore, Riquelme et al. (2018) created the "wheel dataset", which lets us measure performance under sparse reward conditions. We propose to expand the wheel dataset by a parameter that controls the representational complexity of the task, and use it to perform an ablation study on a set of NK predictive distributions (He et al., 2020; Lee et al., 2020) with stochastic policies. We remark that our approach provides a general framework for evaluating sequential decision algorithms.

Our contributions can be summarized as follows:

- We utilize recent results on equivalence between NKs and deep neural networks to derive practical bandit algorithms.
- We show that NK bandits achieve state-of-the-art performance on complex structured data.
- We propose an empirical framework that decouples evaluation of representation learning and exploration in sequential decision processes.

Within this framework:

- We evaluate the most common NK predictive distributions and bandit policies and assess their impact on the key aspects of bandits — exploration and exploitation.
- We analyze the efficacy of a Student's t-process (TP), as proposed by Shah et al. (2013); Tracey & Wolpert (2018), in improving exploration in NK bandits.

We make our work fully reproducible by providing the implementation of our algorithm and the experimental benchmark. [1] The explanation of the mathematical notation can be found in Sec. A.

## 2 BACKGROUND

In this section we provide a necessary background on NK bandits from the perspective of Bayesian inference. Our approach focuses heavily on the posterior predictive derivation and the Bayesian interpretation of predictive distributions obtained as ensembles of inite-width deep neural networks. We build on material related to neural bandits, NKs, and the posterior predictive. We also point out the unique way in which the methods are combined and utilized in our approach.

### 2.1 CONTEXTUAL BANDIT POLICIES

Contextual multi-armed bandits are probabilistic models that at each round $t \in [T]$ of a sequential decision process receive a set of $k$ arms and a global context $\mathbf{x}_t$. The role of a *policy* $\pi$ is to select an action $a \in [k]$ and observe the associated reward $y_{t,a}$. We name the triplet $(\mathbf{x}_t, a_t, y_{t,a_t})$ an *observation* at time $t$. Observations are stored in a dataset, which can be joint ($\mathcal{D}$) or separate for each arm ($\mathcal{D}_a$). The objective is to minimize the cumulative regret $R_T = \mathbb{E}\big[\sum_{t=1}^{T} \max_a y_{t,a} - \sum_{t=1}^{T} y_{t,a_t}\big]$ or, alternatively, to maximize the cumulative reward $\sum_{t=1}^{T} y_{t,a_t}$.

Thompson sampling (TS) is a policy that operates on the Bayesian principle. The reward estimates for each arm are computed in terms of a posterior predictive distribution $p(y_*|a, D_a, \boldsymbol{\theta}) = \int p(y_*|\boldsymbol{\theta}) p(\boldsymbol{\theta}|D_a) d\boldsymbol{\theta}$. Linear, kernel, and neural TS (Agrawal & Goyal, 2013; Chowdhury & Gopalan, 2017; Riquelme et al., 2018) are all contextual variants of TS, which commonly model

---

[1] Code link temporarily hidden for blind review.

| Predictive GP | $\mu$ | $\sigma^2$ |
|---|---|---|
| **NNGP** | $\mathcal{K}_{x_* X}(\mathcal{K}_{XX} + \gamma\mathbf{I})^{-1}\mathbf{y}$ | $\mathcal{K}_{x_* x_*} - \mathcal{K}_{x_* X}(\mathcal{K}_{XX} + \gamma\mathbf{I})^{-1}\mathcal{K}_{Xx_*}$ |
| **Deep Ensembles** | $\Theta_{\mathbf{x}_* \mathbf{x}}\Theta_{\mathbf{XX}}^{-1}\mathbf{y}$ | $\mathcal{K}_{x_* x_*} + \Theta_{\mathbf{x}_* \mathbf{x}}\Theta_{\mathbf{XX}}^{-1}\mathcal{K}_{XX}\Theta_{\mathbf{XX}}^{-1}\Theta_{\mathbf{Xx}_*} - 2\Theta_{\mathbf{x}_* \mathbf{x}}\Theta_{\mathbf{XX}}^{-1}\mathcal{K}_{Xx_*}$ |
| **Randomized Prior** | $\Theta_{\mathbf{x}_* \mathbf{x}}(\Theta_{\mathbf{XX}} + \gamma\mathbf{I})^{-1}\mathbf{y}$ | $\mathcal{K}_{x_* x_*} + \Theta_{\mathbf{x}_* \mathbf{x}}(\Theta_{\mathbf{XX}} + \gamma\mathbf{I})^{-1}\mathcal{K}_{XX}(\Theta_{\mathbf{XX}} + \gamma\mathbf{I})^{-1}\Theta_{\mathbf{Xx}_*}$ |
| | | $-2\Theta_{\mathbf{x}_* \mathbf{x}}(\Theta_{\mathbf{XX}} + \gamma\mathbf{I})^{-1}\mathcal{K}_{Xx_*}$ |
| **NTKGP** | $\Theta_{\mathbf{x}_* \mathbf{x}}(\Theta_{\mathbf{XX}} + \gamma\mathbf{I})^{-1}\mathbf{y}$ | $\Theta_{\mathbf{x}_* \mathbf{x}_*} - \Theta_{\mathbf{x}_* \mathbf{x}}(\Theta_{\mathbf{XX}} + \gamma\mathbf{I})^{-1}\Theta_{\mathbf{Xx}_*}$ |

Table 1: NK predictive distributions. Table reproduced from (He et al., 2020).

the rewards in terms of GP feature regression $y = \mathbf{\Phi\theta} + \epsilon$, where $\mathbf{\Phi} = \phi(\mathbf{X})$ and $\epsilon \sim \mathcal{N}(0, \sigma_y)$. Assuming the prior for each parameter to be $\theta \sim \mathcal{N}(0, \sigma_\theta)$, the analytical solution to the model is $\hat{\mathbf{\theta}} = (\mathbf{\Phi}^T\mathbf{\Phi} + \gamma\mathbf{I})^{-1}\mathbf{\Phi}^T\mathbf{y}$, where $\gamma$ is a regularizer, and a covariance matrix $\mathbf{\Sigma_{XX}} = \mathbf{\Phi}^T\mathbf{\Phi}$. Together $\hat{\mathbf{\theta}}$ and $\mathbf{\Sigma_{XX}}$ parametrize the marginal predictive distribution used to obtain reward estimates $p_{a,t}$.

If the number of parameters $p$ is larger than the number of samples $n$, it is advantageous to apply the kernel trick to reduce the amount of computation, that is to use the identity $(\mathbf{\Phi}^T\mathbf{\Phi} + \gamma\mathbf{I})^{-1}\mathbf{\Phi}^T = \mathbf{\Phi}^T(\mathbf{\Phi}\mathbf{\Phi}^T + \gamma\mathbf{I})^{-1}$ to convert the $p \times p$ covariance matrix $\mathbf{\Sigma_{XX}} = \phi(\mathbf{X})^T\phi(\mathbf{X})$ into a smaller $n \times n$ kernel matrix $\mathbf{K_{XX}} = \mathbf{K}(\mathbf{X}, \mathbf{X}) = \phi(\mathbf{X})\phi(\mathbf{X})^T$. This results in a posterior predictive distribution:

$$p(y_*|\mathbf{x}_*, D) = \mathcal{N}\left(\mathbf{K_{x_* X}}(\mathbf{K_{XX}} + \gamma\mathbf{I})^{-1}\mathbf{y}, \quad \sigma_\theta^2(\mathbf{K_{x_* x_*}} - \mathbf{K_{x_* X}}(\mathbf{K_{XX}} + \gamma\mathbf{I})^{-1}\mathbf{K_{Xx_*}})\right). \quad (1)$$

This distribution was originally applied to bandits by Srinivas et al. (2010); Valko et al. (2013).

A contextual bandit can be either joint or disjoint. A joint model (e.g. Chu et al. (2011)) uses a single kernel and makes predictions based on action-specific contexts, while a disjoint model uses a separate kernel per action, which is more suited for a global context scenario.

While TS offers an explicit Bayesian interpretation, other stochastic policies, like upper confidence bound (UCB) (Auer et al., 2002), can take advantage of the same inference process, and use the moments of the predictive distribution to compute the reward estimates.

**Neural bandits.** Neural bandits are a special case of nonlinear contextual bandits, in which the function $f(x)$ is represented by a neural network. The neural-greedy (e.g. Allesiardo et al. (2014)) approach uses predictions of a NN directly to choose the next action. It is considered a naive approach, tantamount to exploitation in feature space. Neural-linear (Riquelme et al., 2018) utilizes probabilistic regression with $\phi(x)$ representing its penultimate layer features. Nabati et al. (2021) proposed a limited memory version of neural-linear (LiM2), which uses likelihood matching to propagate information about the predictive parameters. This approach is in contrast to storing the data and re-training the network at each iteration in order to prevent catastrophic forgetting.

## 2.2 NEURAL KERNEL DISTRIBUTIONS

We define a neural kernel (NK) to be any compositional kernel (Cho & Saul, 2009; Daniely et al., 2017) that mimics a state or behavior of a neural network. We consider two recently proposed kernels: the neural network Gaussian process kernel (NNGP) (Lee et al., 2018), $\mathcal{K}$, and the neural tangent kernel (NTK) (Jacot et al., 2020), $\Theta$. Both were shown to model the behavior of an infinitely wide network, and allow for training by means of Bayesian inference. He et al. (2020) distinguished several NK predictive distributions, corresponding to common approaches in Bayesian deep learning (Tab. 1).

The NNGP kernel models the network's output covariance at random initialization. It can be composed recursively, such that the kernel of each subsequent layer $\mathcal{K}^{(l)}$ denotes the expected covariance between the outputs of the current layer, for inputs sampled from a GP induced by a kernel of the previous layer $\mathcal{K}^{(l-1)}$. The output of the last layer $\mathcal{K} = \mathcal{K}^{(L)}$ induces a prior GP of the Bayesian regression model and the NNGP posterior predictive can be obtained by inference (Tab. 1).

Jacot et al. (2020) showed that the dynamics of gradient descent optimization, when examined from functional perspective, can be captured by another kernel, NTK, with a general form $\nabla_{\mathbf{\theta}} f(\mathbf{X})\nabla_{\mathbf{\theta}} f(\mathbf{X})^T$, and with the corresponding feature space covariance $\mathbf{Z} = \nabla_{\mathbf{\theta}} f(\mathbf{X})^T\nabla_{\mathbf{\theta}} f(\mathbf{X})$, referred to as the neural tangent feature matrix (NTF). In practice we compute NTK recursively

Figure 1: Noisy wheel dataset. The coordinates represent input features, and the colors represent classes. $\varepsilon$ regulates the complexity of the problem linearly w.r.t. the common baselines (Sec. B.3).

$\Theta = \sum_{l=1}^{L+1} \left( \mathcal{K}^{(l-1)} \prod_{l'=l}^{L+1} \dot{\mathcal{K}}^{(l')} \right)$, with dot denoting a gradient. For networks with ReLU activations both NNGP and NTK become variants of the arc-cosine kernel (Cho & Saul, 2009).

In the infinite width limit the parameters change so little during the training ($O(1/\sqrt{p})$, where $p$ is the number of parameters) that the network can be approximated with a first order Taylor expansion about the initial parameters (Lee et al., 2020). That approximation is referred to as a "linearized network". By putting a Gaussian prior of $\mathcal{N}(\mathbf{0}; \mathcal{K})$ on the functional form of the neural network, the linearized NTK model results in a GP predictive distribution with moments outlined in Tab. 1 - Deep Ensembles. This formulation is given by Lee et al. (2020) (eq 16), who point out, however, that it does not admit the interpretation of a Bayesian *posterior*, yet can still be used as a predictive distribution. When a regularization term is added to the GP, He et al. (2020) showed that this approach also corresponds to networks with randomized priors (Osband et al., 2018) (Tab. 1). He et al. (2020) further attempted to remedy the lack of admittance as a posterior by adding the missing variance of the parameters of all layers except the last one, to the linearized network formulation, which resulted in the *NTKGP* distribution (Tab. 1).

## 3 METHODS

Our extensions are designed to provide insights to the behavior of the NK bandits along the axes of Bayesian inference and exploration-exploitation trade-off. We begin this section with a proposal to expand the wheel dataset in a way that lets us decouple representation learning from exploration, and provide grounds for analysis of differences among the posterior predictive distributions. We then discuss details of the main algorithm, with the emphasis on implementation considerations and the choice of posterior predictive. We close by discussing the conjugate prior approach to the posterior derivation.

### 3.1 WHEEL DATASET AND ITS EXTENSION

In the "wheel" dataset the data is organized in a 2-D Euclidean space, such that $\forall x : ||x|| \leq 1$, and classes are separated by inner circles that let us smoothly regulate the class proportions (e.g. Li & Abu-Mostafa (2006)). Riquelme et al. (2018) introduced a bandit variant of the wheel dataset, which can be seen in the far left subplot of Fig. 1. The parameter $\delta$ controls the proportion of the inner circle to the peripheral classes, which are given the highest rewards. By increasing $\delta$ we ask the algorithm to explore more in order to find the high reward classes. Our reward distribution follows directly from Riquelme et al. (2018) and is detailed in Sec. B.1.

We propose to extend the wheel dataset by an orthogonal parameter that regulates the representation learning difficulty. The "difficulty" is established in terms of data complexity, following a body of work on measuring the complexity of machine learning problems (Li & Abu-Mostafa, 2006; Lorena et al., 2019). In order to generate the data, we morph the original wheel dataset such that it predictably decays the performance of a specfied algorithm. In particular, we propagate the wheel dataset through a fully-connected MLP, initialized with random weights, $w_{ij}^{(l)} \sim \mathcal{N}(0, \frac{\varepsilon}{\sqrt{d^{(l)}}})$, where $d^{(l)}$ is the number of input dimensions to a particular layer, and $\varepsilon$ is a scalar. In the experiments we use ReLU activation functions and $L = 5$ fully-connected layers. The newly introduced "noise parameter" $\varepsilon$ causes the dataset to be warped nonlinearly and injects a moderate level of noise around samples. The effect of increasing $\varepsilon$ can be seen from left to right in Fig. 1. By varying $\delta$ and $\varepsilon$ we create tasks of varying exploration and representation learning difficulty. More details on how the morphing influences complexity can be found in Sec. B.3 and in Fig. 6.

## 3.2 Neural kernel bandits

---

**Algorithm 1** Neural kernel bandit with randomized prior posterior predictive distribution (disjoint)

---

**Require:** Number of arms $k$, number of rounds $T$, kernel regularization parameter $\gamma$, exploration parameter $\eta$, NNGP kernel function $\mathcal{K}(\cdot,\cdot)$, neural tangent kernel function $\Theta(\cdot,\cdot)$, initial number of steps $\iota$, policy $\pi$

1: Play each arm sequentially for $\iota$ steps to accumulate data $\mathbf{X}_a \in \mathbb{R}^{\iota \times d}, \mathbf{y}_a \in \mathbb{R}^\iota \ \forall a$
2: **for** round $t = 1, 2, ..., T$ **do**
3:      Observe context $\mathbf{x}_t$
4:      **for** arm $a = 1, 2, ..., k$ **do**
5:          $\mathcal{K}_{a,t} \leftarrow \mathcal{K}(\mathbf{X}_a, \mathbf{X}_a)$
6:          $\mathbf{\Theta}_{a,t} \leftarrow \Theta(\mathbf{X}_a, \mathbf{X}_a) + \gamma \mathbf{I}$
7:          // Calculate predictive distribution moments
8:          $\mu_{a,t} \leftarrow \mathbf{\Theta}_{\mathbf{x_t}\mathbf{x_a}} \mathbf{\Theta}_{a,t}^{-1} \mathbf{y}_a$
9:          $\sigma_{a,t}^2 \leftarrow \mathcal{K}_{\mathbf{x_t}\mathbf{x_t}} + \mathbf{\Theta}_{\mathbf{x_t}\mathbf{X_a}} \mathbf{\Theta}_{a,t}^{-1} \mathcal{K}_{a,t} \mathbf{\Theta}_{a,t}^{-1} \mathbf{\Theta}_{\mathbf{X_a}\mathbf{x_t}} - 2\mathbf{\Theta}_{\mathbf{x_t}\mathbf{X_a}} \mathbf{\Theta}_{a,t}^{-1} \mathcal{K}_{\mathbf{X_a}\mathbf{x_t}}$
10:          **if** $\pi$ is UCB **then**
11:              $p_{a,t} \leftarrow \mu_{a,t} + \frac{\eta}{\gamma^{1/2}} \sigma_{a,t}$
12:          **else if** $\pi$ is TS **then**
13:              $p_{a,t} \sim \mathcal{N}(\mu_{a,t}, \frac{\eta}{\gamma}\sigma_{a,t}^2)$
14:          **end if**
15:      **end for**
16:      Choose $a_t \leftarrow \arg\max_a p_{a,t}$ and obtain reward $y_t$
17:      Update $(\mathbf{X}_{a_t}, \mathbf{y}_{a_t})$ with $(\mathbf{x}_t, y_t)$
18: **end for**

---

Our algorithm (Alg. 1) is built using a similar structure to classic kernel and GP bandits (Chowdhury & Gopalan, 2017; Srinivas et al., 2010; Valko et al., 2013). We compute GP moments $\mu_{a,t}$ and $\sigma_{a,t}$ according to Tab. 1, by modifying the definitions of lines 8 and 9. As an example, in Alg. 1 we use the randomized prior approach. We provide both UCB and TS variants of our algorithm. In each round an arm is chosen based on the highest estimate of the reward $p_{a,t}$. We use the disjoint approach as our primary model, which means that the data $(\mathbf{X}_a, \mathbf{y}_a)$ and kernels (NNGP $\mathcal{K}_{a,t}$ and NTK $\mathbf{\Theta}_{a,t}$) are collected and computed separately for each arm $a$. The disjoint strategy preserves memory, while ensuring that better performing arms, chosen more frequently, have larger associated datasets and thus more certainty over time. This approach is very much in line with classic optimism in face of uncertainty (OFU) methods. A common alternative is the joint model, which, in the case of GP, uses a single kernel and produces separate posteriors, albeit with a common uncertainty estimate, for each arm at prediction time. Using NKs in this manner for multi-class problems is common in the NNGP/NTK literature (e.g. Arora et al. (2019b); Novak et al. (2019)). However, the approach is not feasible in the bandit setting as it assumes access to rewards associated with all the arms. A common solution (e.g. Zhou et al. (2020)) is to use zero-padding to create a separate context vector for each arm $\mathbf{x}_a = [\mathbf{0}, \ldots, \mathbf{0}, \mathbf{x}_{orig}, \mathbf{0}, \ldots, \mathbf{0}]$. In the feature space this has an effect of splitting the model's parameters per arm yielding separate uncertainty estimates.

## 3.3 Student's t-process

While our GP predictive distributions represent the posterior predictives with the fixed observation noise $\sigma_y$, we also analyze the behavior of the algorithm in conjunction with the joint prior. Linear and neural-linear models were shown by Nabati et al. (2021) and Riquelme et al. (2018) to benefit from applying a joint conjugate prior to both parameters of the likelihood, resulting in a Normal-inverse-Gamma distribution $p(\boldsymbol{\theta}, \sigma_y^2) = p(\boldsymbol{\theta}|\sigma_y^2)p(\sigma_y^2)$. When $\sigma_y$ is not fixed, it continues to evolve with the data, which results in more accurate uncertainty estimates over time. The joint prior inference can be performed either by (1) a two-stage sampling process, where parameters are sampled individually, and then used in the regression model, or (2) by a full predictive distribution derivation. We use the first approach in linear and neural-linear bandits. When using kernels, the parameters

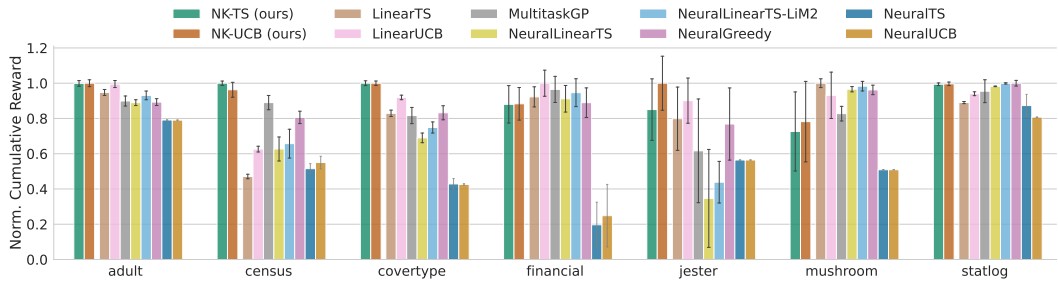

Figure 2: Final normalized cumulative rewards (y-axis) of NK bandits compared against all other methods after 5,000 steps rollout on the respective UCI datasets (x-axis). Each bar shows the mean and standard deviation over 10 rollouts.

cannot be sampled, and require a full derivation, which results in a Student's t-process (TP):

$$p(y_*|\mathbf{x}_*, D) = \mathcal{T}\left(\mathbf{K}_{\mathbf{x}_*\mathbf{x}}(\mathbf{K}_{\mathbf{X}\mathbf{X}} - \gamma\mathbf{I})^{-1}\mathbf{y}, \quad \frac{\nu + \mathbf{y}^T(\mathbf{K}_{\mathbf{X}\mathbf{X}} - \gamma\mathbf{I})^{-1}\mathbf{y} - 2}{\nu + n - 2}\widehat{\mathbf{K}}_{\mathcal{T}}, \quad \nu + n\right),$$

$$\widehat{\mathbf{K}}_{\mathcal{T}} = \frac{\nu - 2}{\nu}\left(\mathbf{K}_{\mathbf{x}_*\mathbf{x}_*} - \mathbf{K}_{\mathbf{x}_*\mathbf{x}}(\mathbf{K}_{\mathbf{X}\mathbf{X}} - \gamma\mathbf{I})^{-1}\mathbf{K}_{\mathbf{X}\mathbf{x}_*}\right). \tag{2}$$

The TP is related to the GP by having the same mean, but a scaled covariance, by a factor of $\frac{\nu}{\nu-2}$. $\nu$ is known as "degrees of freedom", and controls the kurtosis. The lower $\nu$ values result in heavier tails, and thus lead to more aggressive exploration. TP approaches GP as $\nu \to \infty$, and can therefore be interpreted as a generalization of the GP (Tracey & Wolpert, 2018). Correspondence to GP allows us to use the same NK-based distribution parameters in an unchanged form. To introduce finer control over performance, we make use of a regularizer $\gamma$, although we note that it does not have the same interpretation as in the GP. The key advantage of TP over GP is that its variance depends on both input and output data, rather than just inputs. The output variance is introduced through the $\mathbf{y}^T(\mathbf{K}_{\mathbf{X}\mathbf{X}} - \gamma\mathbf{I})^{-1}\mathbf{y}$ term, which represents the data complexity. This term will drift from $n$ the more the outputs deviate from the GP modelling assumption (Tracey & Wolpert, 2018).

## 4 EXPERIMENTS

While NK bandits were proposed in the past, the work presented by Zhou et al. (2020); Zhang et al. (2020) focuses mainly on theoretical analysis and offers only a limited empirical evaluation of an approach that can be considered a feature space variant of NTKGP. We conduct experiments to show that NKs provide a competitive approach in bandit settings. We begin by evaluating our method on a Bayesian bandit benchmark (Riquelme et al., 2018) and show that it achieves on-par or better performance than state-of-the-art baselines on most of the tested UCI datasets. We conclude that the method should be considered by the community as one of the standard approaches for bandits with nonlinear reward models. Next, we utilize the noisy wheel dataset to assess the impact of utilizing NK predictive distributions to accurately trade off exploration and exploitation. While no approach can adapt to all settings, we observe that NTKGP provides a significant improvement in sparse environments requiring large exploration. In addition to the main results, we assess the ability of the noisy wheel dataset to model complexity, and measure the implications of practical implementation considerations — training frequency and model partitioning (Sec. B.4, Sec. B.3, and Sec. B.5).

### 4.1 NK BANDITS OUTPERFORM NEURAL-LINEAR AND NTF BANDITS ON COMPLEX DATA

We tested the performance of NK bandits on a set of UCI datasets (Dua & Graff, 2019) used in the Bayesian bandit benchmark (Riquelme et al., 2018). We compared our method against the state-of-the-art neural-linear (Riquelme et al., 2018) and neural-linear LiM2 (Nabati et al., 2021) bandits, following their experimental setup. We report the mean and standard deviation of the cumulative reward over 10 runs. We included important baselines: linear TS / UCB and multitask GP, due to their notable performance on simpler datasets. According to a common practice in the field (Riquelme et al., 2018; Nabati et al., 2021), we did not tune the hyperparameters of any algorithm. Unlike

in supervised learning, bandit benchmarks typically do not provide a separate test set. Instead, we pre-set the hyperparameters to values established by Riquelme et al. (2018), whenever applicable. Due to differences in the NN architecture between Riquelme et al. (2018) and Nabati et al. (2021), in Tab. 3 we report the results for both settings. We used the neural-tangents library (Novak et al., 2019) to obtain the NTK, NNGP kernel, and the mean and covariance of the predictive distributions. We computed the NKs based on a two-layer fully-connected architecture with ReLU activations and regularizer $\gamma = 0.2$. We chose the randomized prior GP (He et al., 2020; Lee et al., 2020; Osband et al., 2018) as our preliminary experiments showed it to perform best on the majority of the selected UCI datasets (not shown in the paper). In light of the findings presented in Sec. 4.2 and the assessment by Riquelme et al. (2018) we suspect that deep ensemble and randomized prior distributions are more suitable for datasets like UCI, that require fewer exploration steps. We tested both UCB and TS policies with the exploration parameter $\eta$ set to 0.1. As established by Nabati et al. (2021), normalized cumulative rewards are computed w.r.t. naïve uniform sampling and the best algorithm for each dataset: $\text{norm\_cum\_rew}_{\text{alg}} = \frac{\text{cum\_rew}_{\text{alg}} - \text{cum\_rew}_{\text{uniform}}}{\text{cum\_rew}_{\text{best}} - \text{cum\_rew}_{\text{uniform}}}$.

The results (Fig. 2) show that NK bandits constitute a competitive approach, achieving state-of-the-art results on most datasets. The improvement is most apparent on datasets with high non-linear entanglement and large number of features ($> 50$). Interestingly, we attain the highest performance boost on the covertype dataset, which is typically considered the hardest. We attribute the increase to accurate exploitation, achieved by application of NTK to nonlinear structured data problems (Arora et al., 2019b), and exploration, achieved through the appropriate choice of the predictive distribution. For linear datasets, linear methods tend to outperform non-linear ones, including ours. The same trend was noted by Riquelme et al. (2018) w.r.t. previous methods.

We recognize that scalability is a current limitation of our method. Computing NTK requires large computational resources. In Sec. B we show two avenues for scaling up, with relatively low impact on performance. Disjoint arm treatment (Sec. B.5) provides an opportunity for distributed computing, while lowering the training frequency (Sec. B.4) allows for considerable computational gain with minimal performance drop. We also note that current research directions provide promising avenues for significant scaling of our approach in the near future (Zandieh et al., 2021).

## 4.2 DECOUPLING REPRESENTATION LEARNING FROM UNCERTAINTY ESTIMATION

Decoupling representation learning from uncertainty estimation allows us to study each factor separately and understand their impact on the overall performance of bandits. In the following experiment we utilize the noisy wheel dataset to measure the performance of TS and UCB policies combined with NK-induced TPs and GPs (Tab. 1), w.r.t. the varying factors of complexity $\varepsilon$ and the need to explore $\delta$. We compared all methods by measuring their *peripheral accuracy*, i.e. the percentage of correctly chosen actions on the periphery of the wheel: $pacc = \sum_i I[a_i^* \neq a^0] I[a_i = a_i^*] / \sum_i [a_i^* \neq a^0]$. We evaluated each algorithm on a grid of $(\varepsilon, \delta)$ pairs, where $\varepsilon$'s were spaced linearly, due to their linear impact on performance (see Sec. B.3), and $\delta$'s were spaced following Riquelme et al. (2018). The grid was computed separately for each {distribution,policy} pair. Each method was run for 5,000 steps and the results were averaged over 10 runs. Due to the large number of runs, we updated the model every 20 steps to speed up the computation (see Sec. B.4). All experiments were performed using a disjoint model with exploration parameter $\eta = 0.1$. The distribution were regularized with $\gamma = 0.2$. TP was initialized with $\nu = 12$. The results are presented in Fig. 3.

While there is no single method that achieves the highest score in all circumstances, in average the NTKGP distribution provides the best performance w.r.t. the exploration-exploitation trade-off. All predictive distributions and policies exhibit good performance close to the original wheel problem with $\delta = 0.5$ (Fig. 3; top left corner of each grid), and gradually worsen as problems become more complicated on both axes. The algorithms eventually fail when $\delta = 0.99$. We argue that a good bandit method should be able to adapt and sustain its performance longer as the problem becomes more complex and requires more exploration. In Fig. 3 we can see that NTKGP clearly dominates on harder problems, especially in combination with Thompson sampling. This is indicated by almost uniformly higher scores (3rd column, 1st and 2nd row) and can be further verified by observing the histograms (rhs). The result is in contrast with our finding that the randomized prior distribution performed best on the UCI datasets. We believe that the difference is due to the reduced need to explore in the UCI tasks (Riquelme et al., 2018). Fig. 3 (top histograms) and Fig. 4 further confirm that intuition by showing little distinction in performance when the need for exploration is small (low $\epsilon$).

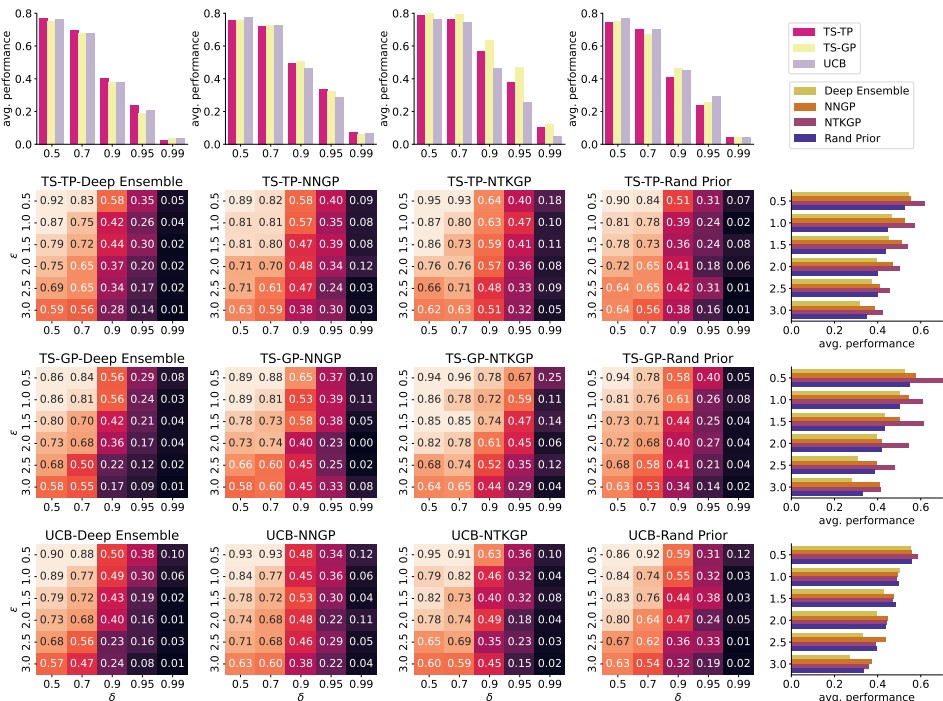

Figure 3: Exploration and exploitation of NK predictive distribution (columns) in combination with bandit policies (rows). Grid cells represent peripheral accuracies. Histograms represent mean results (by method) over columns or rows respectively.

NTKGP was shown to provide more accurate out-of-distribution (OOD) and occasionally worse in-distribution predictions (He et al., 2020). In bandits, the improved OOD performance is crucial, due to its effect on initial exploration. After the uncertainty converges, it plays a significantly lesser role in overall performance. NTKGP was also shown by He et al. (2020) to make more "conservative" decisions on samples it is uncertain about. In the ensemble settings, studied by He et al. (2020), this means that the algorithm utilizes uncertainty as its measure of confidence, and tends to take more confident, less extreme actions. In bandits we use the larger uncertainty to produce an opposite effect, and decide to sample the uncertain regions more frequently, which facilitates exploration.

NNGP often performs better than deep ensembles and randomized prior distributions, but worse than NTKGP. It is interesting to see that a method based purely on the NNGP kernel outperforms more sophisticated distributions that combine both kernels. While deep ensemble and randomized prior approaches are well founded in representing NNs after training (Lee et al., 2018), NNGP distribution operates on a randomly initialized network. However, NNGP and NTKGP are fully Bayesian approaches, while deep ensemble and randomized priors are not. The results suggest that the Bayesian interpretation may be advantageous in uncertainty estimation.

The deep ensemble distribution performs much worse than other methods, while not being the simplest to compute. This shows a large impact of the regularizer on the overall performance. Interestingly, also the disparity in performance between deep ensemble and all other methods grows significantly as the data complexity increases. This disparity is less noticeable when changing the exploration parameter delta (compare with Fig. 5).

Fig. 3 shows that UCB has less variance in performance between the distributions. We attribute the lower variance to the fewer number of exploration steps that UCB performs over the rollout. Fewer exploration steps mean less reliance on uncertainty estimation, and thus smaller distinction between the tested predictive distributions. While we tried a limited sensitivity analysis for setting the $\eta$ parameter to bring TS and UCB to similar performance, the results were inconclusive. We set $\eta = 0.1$ for both methods and leave a more extensive assessment for future work.

## 5   RELATED WORK

The work of (Chu et al., 2011; Li et al., 2010) on linear UCB and (Agrawal & Goyal, 2013) on linear TS motivated an abundance of research on regression bandits. GP-UCB (Srinivas et al., 2010) and GP-TS (Chowdhury & Gopalan, 2017) are special cases of kernel bandits (Valko et al., 2013), in which the ridge regularizer is set to Gaussian noise to form a GP. In practical applications, bandits are traditionally combined with well-known kernels, like linear, RBF or Matérn (Rasmussen & Williams, 2006). More recently a new family of neural kernels (NK) has emerged, which directly approximates the behavior of deep neural networks (NN) (Cho & Saul, 2009; Daniely et al., 2017). The neural network Gaussian process (NNGP) kernel is derived from a forward pass of a NN, and corresponds to a NN with randomly initialized weights (Lee et al., 2018). The finding of a direct GP correspondence was later followed by establishing a similar relation for linearized NNs and the neural tangent kernel (NTK), where NTK represents a network's dynamics during gradient descent training (Arora et al., 2019a; He et al., 2020; Jacot et al., 2020; Lee et al., 2020). A review of all recent NK-GP models, with connections to other existing uncertainty models, like deep ensembles and randomized priors (Osband et al., 2018), can be found in (He et al., 2020).

NeuralUCB (Zhou et al., 2020) and NeuralTS (Zhang et al., 2020) are two related neural bandit policies that can be considered feature space variants of NK bandits. They have an NTKGP posterior (He et al., 2020), whose moments are defined by a standard NN estimate and the neural tangent features (NTF) covariance (Sec. 2.2). These algorithms were shown to achieve a regret bound comparable to kernel bandits with the effective dimension induced by a corresponding NTK (Zhou et al., 2020; Zhang et al., 2020). Yet, they were shown to underperform in practical applications (Zhou et al., 2020; Zhang et al., 2020; Nabati et al., 2021), which was linked to overparametrization and poor covariance approximation (Xu et al., 2020). Our work directly addresses these limitations.

The extended Kalman filter (EKF) provides another fully Bayesian treatment of all model parameters that was also applied to neural bandits (Duran-Martin et al., 2021). The EKF approach was shown to be particularly suitable to online learning, as it operates on constant memory. In practice, however, the approach relies on dimensionality reduction to approximate parameters of a linearized network. The NK approach (NTK and NNGP), on the other hand, applies the law of large numbers and central limit theorem to construct an exact predictive distribution recursively from an infinite network, which is also linearized, but the method does not rely on further approximation. In this work we are interested in the NK approach, with the goal of testing the performance of neural bandits in the presence of exact uncertainty estimates.

## 6   CONCLUSIONS

In this paper we evaluated NK bandits on a popular benchmark, comprising real-world (UCI) and synthetic (wheel) datasets. We proposed to guide the bandit policy by Student's t or Gaussian processes corresponding to common classes of infinite neural networks. We showed that our model outperforms the state-of-the-art methods on the majority of tasks. We also note that it performs best when the number of features is small, or when the reward is highly non-linear. In addition, we expanded the wheel dataset (Riquelme et al., 2018) to measure both the ability to explore and to learn complex representations. We used our proposed framework to evaluate the impact of NK predictive distributions on the two key aspects of bandits, exploration and exploitation, and showed that while there is no single distribution applicable to all scenarios, NTKGP in combination with Thompson sampling is most robust to changes in the complexity and the need for exploration.

We consider several directions for further research. First, the limitation of our method is its scalability. While we offered several options with large performance to computation time ratios, we believe our approach can greatly benefit from kernel approximation methods, like Zandieh et al. (2021). We then plan to investigate further the difference between TP and GP in handling the exploration and exploitation trade-off under varied conditions. Finally, our empirical framework could be used to analyze other fully Bayesian neural bandits, e.g. Duran-Martin et al. (2021). It would be also interesting to compare the exploration and representation learning trade-offs for other methods from (Riquelme et al., 2018), especially neural-linear that was said to benefit from decoupling.

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

## A  MATHEMATICAL NOTATION

We denote the training data by $(\mathbf{X}, \mathbf{y}) \in (\mathcal{X}, \mathcal{Y})$, and the test points by $(\mathbf{x}_*, y_*)$. The input data is composed of $(\mathbf{x}_i, y_i)$, where $i \in [n]$, when we talk about a predetermined set. Alternatively the data points can be indexed by time $t \in [T]$ in the context of bandits, where the data is collected sequentially. When a subset of data is associated with a specific action we put it in a subscript, i.e. $(\mathbf{X}_a, \mathbf{y}_a)$. Parameter vectors are denoted by $\boldsymbol{\theta}$ and the reward estimation models by $f : \mathcal{X} \to \mathcal{Y}$. The dimensionality of both the feature vector and the parameter vector is denoted by $p$, i.e. $\boldsymbol{\theta} \in \mathbb{R}^p$ and $\phi(\mathbf{x}) \in \mathbb{R}^p$. In the linear case $\phi(\mathbf{x}) = \mathbf{x}$, so also $\mathbf{x} \in \mathbb{R}^p$. We denote a generic kernel matrix by $\mathbf{K}$ and the associated kernel function by $K(\cdot, \cdot)$, such that $\mathbf{K_{xy}} \equiv K(\mathbf{x}, \mathbf{y})$. We use $\mathbf{x}_*$ for a general test input in the context of Bayesian inference, or $\mathbf{x}_t$ when the test input is given at a specific time $t$. We denote test predictions by $y_*$ or $y_t$. When we talk about the potential rewards from all actions, e.g. in the context of regret, we add a subscript denoting the action, i.e. $y_{t,a_t}$ for the reward associated with the chosen action $a_t$, and $y_{t,a}$ for rewards associated with action $a$.

## B  EXPERIMENTAL SETUP

### B.1  DATASETS

Table 2 provides an summary of the key properties associated with each of the UCI datasets we considered. While Riquelme et al. (2018) and Nabati et al. (2021) used the same datasets, there are subtle differences in the way that they were implemented. Most notably, Nabati et al. (2021) use the "adult" dataset to classify income bracket, while Riquelme et al. (2018) use it to classify occupation. In case of any discrepancies we followed Nabati et al. (2021)'s set up for UCI datasets. In the (noisy) wheel dataset we used the same reward distribution parameters as proposed by Riquelme et al. (2018). The reward for guessing the inner circle class (i.e. where $||x|| \leq \delta$) is $r_{small} \sim \mathcal{N}(1.2; 0.05)$. The reward for the correct peripheral class (sample classified correctly) is $r_{big} \sim \mathcal{N}(50; 0.01)$, while $r_{peripheral} \sim \mathcal{N}(1; 0.05)$.

### B.2  UCI EXPERIMENTATION TIME ANALYSIS

We complement our results by reporting the average (over 10 runs) times per round that our algorithms took w.r.t. the neural-linear approach (Tab. 4). As the times grow with increasing number of collected data points, we report the shortest, median, and the longest average round. Both TS and UCB variants of our method tend to be about 5 times slower than the LiM2 approach, taking around 2 seconds for the longest round. The total times can be further reduced for some applications by performing less frequent updates, as described in Sec. B.4. However, additional study would need to be conducted to check for a similar effect in neural-linear approaches.

### B.3  COMPLEXITY ASSESSMENT OF THE NOISY WHEEL DATASET

SVM performance is one of the standard metrics for empirical assessment of the complexity of a problem (Li & Abu-Mostafa, 2006; Lorena et al., 2019). In Fig. 6 we measured performance of

---

[2]For adult and census datasets the categorical features are subsequently encoded with one hot vectors, resulting in $\sim 80$ features for adult and $\sim 370$ features for census. The numbers vary due to subsampling.

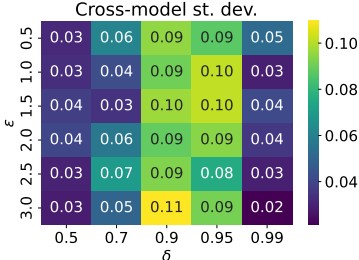

Figure 4: Standard deviations across tested methods (Fig. 3) reveal regions greatest differences in performance, which can be used to rank methods.

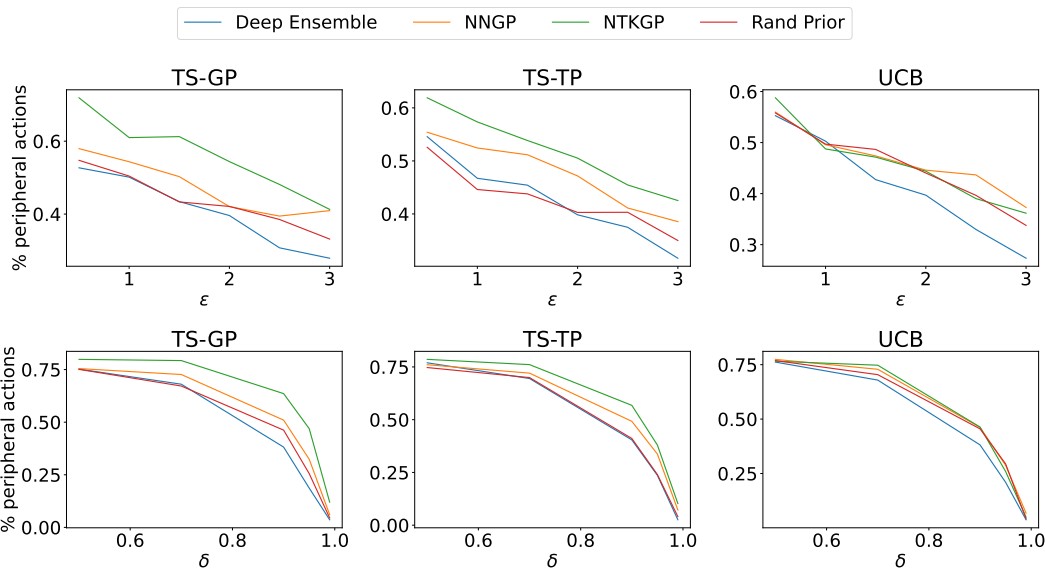

Figure 5: Percentage of correct peripheral actions, averaged across policies and a missing dimension ($\delta$ in top row, and $\epsilon$ in bottom row) for the experiment presented in Fig. 3. The aggregates provide clearer view of the average rank of each NK distribution.

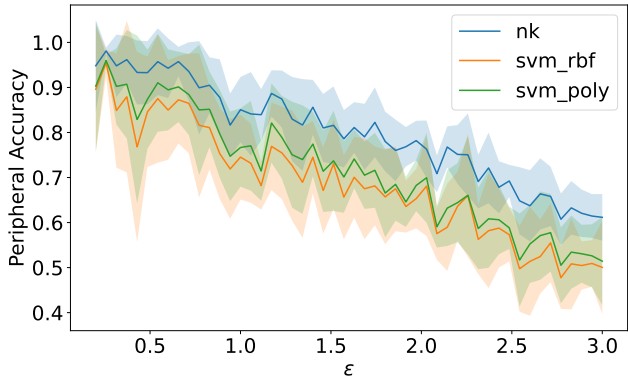

Figure 6: Baseline peripheral accuracy drops linearly as we increase the level of noise $\varepsilon$ in the noisy wheel dataset. The result further supports $\varepsilon$ as a sufficient proxy for problem's complexity.

3 baseline supervised approaches: NK ridge regression, SVM with RBF kernel, and SVM with polynomial kernel. Each method was run 10 times. The results show that the noise parameter increases the difficulty of the problem linearly in the chosen range from the performance perspective when no additional hyperparameter tuning is done.

## B.4 TRAINING FREQUENCY

Frequent updates of complex models in bandit settings put high demand on compute resources. We measure the sensitivity of our method to the number of actions taken between subsequent model updates. We ran our NK bandit for numbers of actions specified in the recent literature (Nabati et al., 2021; Riquelme et al., 2018): $\{1, 5, 20, 100, 400\}$, and report their respective cumulative rewards on the UCI datasets after 5,000 steps. Fig. 7 shows a smooth degradation in performance as the number of actions increases, which is expected. We note, however, that the degradation is relatively small and preserves competitive performance on all tested datasets. As large numbers of actions help to save significant amounts of computational resources, we believe that the sustained performance provides a strong argument for applicability to real world scenarios. It also positions our method as a good candidate for a limited memory approach, which we leave for future work. For some datasets, we observe a significant spike in performance at 5 actions before retraining. The result suggests an additional role the training frequency plays in the exploration-exploitation trade-off. In Fig. 7 we also show the results of the best performing models, other than our method, as reported in Fig. 2. We observe that for some datasets of higher complexity (e.g. covertype), our method's rank w.r.t. other algorithms is not affected, even with large numbers of actions before retraining, which additionally confirms its robustness.

## B.5 JOINT MODEL

Following the zero-padding approach (Sec. 3.2), we compared the performance of joint and disjoint models and noticed a considerable improvement in the last epoch performance for linear and stochastic datasets (financial and mushroom), slight improvement for datasets of medium complexity, and a large deterioration for some of the most difficult datasets — covertype and jester. When examined from a resource usage perspective, however, the joint model underperforms considerably. This result is significant, as most neural bandits use a joint (NeuralUCB, NeuralTS) or partially joint (neural-linear) model. It is important to note that in functional space a disjoint model reduces the overall number of kernel entries, while in the case of parameter space models we observe the opposite effect. We can only create a disjoint model at the expense of introducing additional $(k-1)|\boldsymbol{\theta}|$ number of parameters, which puts a burden on resources. Our result highlights this difference, and can help practitioners assess the impact of choosing joint or disjoint models when using NK bandits.

Table 2: Summary of the UCI datasets used in the experiments.

| Dataset | Description | Distinguishing factor(s) | $d^2$ | $k$ | $\sim n$ | Reward | Context |
|---|---|---|---|---|---|---|---|
| Adult | predict income based on personal information | binary classification | 13 | 2 | 50k | binary | categorical / integer |
| Census | predict occupation based on personal information | multivariate classification; large number of features | 67 | 9 | 250k | binary | categorical / integer |
| Covertype | type of forest coverage in various areas | multivariate classification; requires nonlinear models (Riquelme et al., 2018) | 54 | 7 | 150k | binary | categorical / integer |
| Financial | stock prices from NYSE and NASDAQ | linear problem; easy to over-explore | 21 | 8 | 4k | continuous | continuous |
| Jester | joke recommender system | inputs and outputs in the same domain | 32 | 8 | 20k | continuous | continuous |
| Mushroom | poisonous vs. safe mushrooms | imbalanced stochastic rewards; captures aspects of classification and regression | 117 | 2 | 10k | continuous | categorical |
| Statlog | space shuttle flight indicators | multivariate classification; tests exploration; one arm optimal 80% of the time; requires nonlinear models (Riquelme et al., 2018) | 9 | 7 | 45k | binary | integer |

Table 3: Neural-linear methods compared over two significant hyperparameter settings.

| | adult | census | covertype | financial | jester | mushroom | statlog |
|---|---|---|---|---|---|---|---|
| NK-TS $L = 1$ $\gamma = 0.2$ | **4119 ± 16** | 3152 ± 41 | 3402 ± 41 | 4407 ± 344 | 16779 ± 1234 | 4052 ± 4974 | 4720 ± 132 |
| NK-TS $L = 2$ $\gamma = 0.2$ | 4113 ± 19 | **3186 ± 29** | **3441 ± 36** | 4162 ± 276 | **16856 ± 1212** | 2016 ± 3677 | 4820 ± 32 |
| NeuralLinearTS $L = 1$ $n_l = 50$ | 3970 ± 36 | 2557 ± 54 | 2698 ± 76 | 4358 ± 363 | 12552 ± 1875 | **11111 ± 336** | 4771 ± 13 |
| NeuralLinearTS $L = 2$ $n_l = 100$ | 3927 ± 26 | 2223 ± 181 | 2588 ± 74 | 4196 ± 350 | 11542 ± 2365 | 10516 ± 345 | 4801 ± 7 |
| NeuralLinearTS-LiM2 $L = 1$ $n_l = 50$ | 4044 ± 17 | 2726 ± 36 | 2745 ± 141 | **4485 ± 360** | 14333 ± 1339 | 10962 ± 1070 | 4816 ± 66 |
| NeuralLinearTS-LiM2 $L = 2$ $n_l = 100$ | 3993 ± 39 | 2304 ± 217 | 2748 ± 85 | 4357 ± 366 | 12325 ± 1005 | 10915 ± 631 | **4872 ± 14** |

Table 4: Times per epoch [sec] (min / median / max) rounded to 2 significant figures.

| | adult | census | covertype | financial | jester | mushroom | statlog |
|---|---|---|---|---|---|---|---|
| NTK-TS | 0.0 / 0.8 / 2.25 | 0.0 / 0.07 / 1.93 | 0.0 / 0.64 / 2.47 | 0.0 / 0.02 / 1.79 | 0.0 / 0.48 / 1.62 | 0.0 / 0.82 / 2.66 | 0.0 / 0.85 / 2.42 |
| NTK-UCB | 0.0 / 0.8 / 1.85 | 0.0 / 0.07 / 2.04 | 0.0 / 0.71 / 2.01 | 0.0 / 0.02 / 2.04 | 0.0 / 0.44 / 3.15 | 0.0 / 0.82 / 4.81 | 0.0 / 0.85 / 2.81 |
| JointNTK-TS | 0.67 / 0.9 / 5.68 | 0.0 / 2.0 / 5.5 | 0.0 / 0.97 / 2.43 | 0.0 / 0.83 / 2.51 | 0.0 / 0.92 / 2.87 | 0.68 / 0.94 / 2.47 | 0.0 / 0.83 / 4.13 |
| JointNTK-UCB | 0.67 / 0.91 / 3.16 | 0.0 / 2.0 / 5.49 | 0.0 / 0.97 / 3.56 | 0.0 / 0.83 / 2.27 | 0.0 / 0.89 / 2.38 | 0.67 / 0.94 / 2.45 | 0.0 / 0.81 / 2.82 |
| NeuralLinearTS | 0.0 / 0.0 / 1.2 | 0.0 / 0.0 / 1.3 | 0.0 / 0.0 / 1.24 | 0.0 / 0.0 / 0.96 | 0.0 / 0.0 / 1.22 | 0.0 / 0.0 / 1.19 | 0.0 / 0.0 / 1.21 |
| NeuralLinearTS-LiM2 | 0.02 / 0.03 / 0.33 | 0.04 / 0.1 / 0.36 | 0.03 / 0.06 / 0.36 | 0.04 / 0.08 / 0.38 | 0.03 / 0.09 / 0.37 | 0.02 / 0.03 / 0.32 | 0.03 / 0.08 / 0.36 |

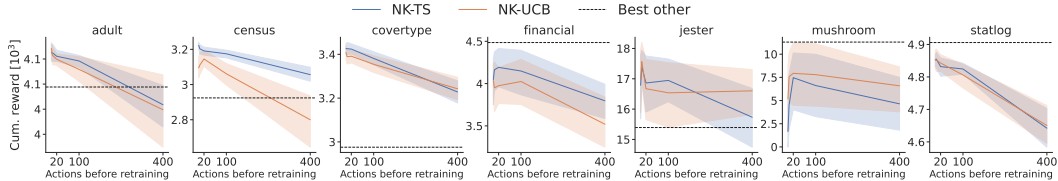

Figure 7: Performance w.r.t. $\{1, 5, 20, 100, 400\}$ actions before retraining.

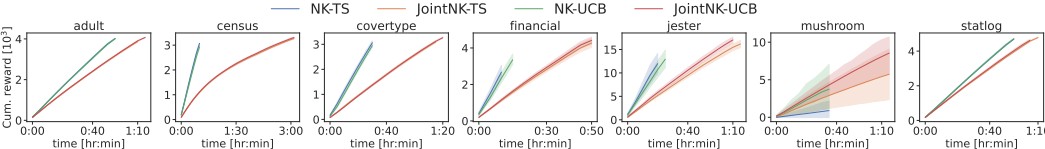

Figure 8: Cumulative rewards w.r.t. running times of a joint and a disjoint variant of our method.

