# OpenReview forum: "Empirical analysis of representation learning and exploration in neural kernel bandits"
_ICLR.cc/2023/Conference — Submitted to ICLR 2023_

### Official Review · Reviewer_uxKz · 2022-10-24

**Confidence:** 3
**Clarity, Quality, Novelty And Reproducibility:** The paper is clearly written and the …
**Correctness:** 4
**Technical Novelty And Significance:** 1
**Empirical Novelty And Significance:** 2
**Recommendation:** 3

**Strength And Weaknesses:**

The background is interesting and the various NN kernel do have some potential for use in GP models for sequential model-based optimisation and bandits.

My main difficulty with this paper however is that, in the end, it appears to come down to a simple experimental comparison of various kernels, some of which happen to derive from various models or analysis of neural networks.  I am not convinced that this suffices to have a significant impact.

**Summary Of The Paper:**

This paper presents am empirical comparison between the performance of various neural kernels used in GPs for contextual bandit optimisation.  Comparisons are done using a modified wheel dataset, and the use of the student t-process to improve exploration in NK bandits is explored.

**Summary Of The Review:**

The paper is well written, but as this boils down to a comparison of kernels in GPs I feel that there is insufficient novelty here to warrant acceptance.

---

### Official Review · Reviewer_r846 · 2022-10-24

**Confidence:** 4
**Correctness:** 3
**Technical Novelty And Significance:** 2
**Empirical Novelty And Significance:** 3
**Recommendation:** 3

**Clarity, Quality, Novelty And Reproducibility:**

Clarity: The paper is well written.

Quality: The empirical evaluations are comprehensive and hence of high quality.

Novelty: The technical novelty may be limited, since the proposed algorithm is a straightforward combination of existing methods.

Reproducibility: The code is submitted for reproducibility.

**Strength And Weaknesses:**

Strengths:
- The empirical insights drawn from the empirical evaluations of this paper are interesting and can be of interest for the neural bandit community and the bandit community in general.
- The proposed benchmark with the additional ability to evaluate the ability of an algorithm to learn representation is also interesting and can be potentially useful for the broader community.

Weaknesses:
- My biggest concern is regarding the comparison with the recent works on neural contextual bandits which explicitly train a neural network to predict the reward, including NeuralUCB and NeuralTS. **Firstly**, Figure 2 shows that NeuralUCB and NeuralTS are consistently outperformed by LinearUCB and LinearTS. This is surprising to me, because in the previous papers on NeuralUCB (Zhou et al., 2020) and NeuralTS (Zhang et al., 2020), they have both also performed experiments using the UCI datasets and had shown that both NeuralUCB and NeuralTS consistently outperform linear bandit algorithms. What's the reason for this discrepancy between your observations and theirs? **Secondly**, it is mentioned in the Introduction that "...NKs have been shown to lack the full representational power of the corresponding NNs...". The same statement has also been made by a few other papers such as NeuralUCB. As a result of this, the use of a neural network to predict the reward (i.e., using an NN as the term $\mu_{a,t}$ on line 8 of Algorithm 1) is important for achieving good performances in neural bandits. Therefore, it is surprising that NeuralUCB and NeuralTS, which indeed use NNs for reward prediction, are significantly outperformed by the proposed algorithm which does not use NNs for reward prediction. Furthermore, in fact, the NeuralUCB paper also proposed another algorithm NeuralUCB0 which does not use NNs for reward prediction but instead simply treats NTK as a kernel in kernelized bandits. In this sense, NeuralUCB0 is similar to the proposed algorithm when NTKGP is used. But in their paper, they have shown that this NeuralUCB0 is consistently outperformed by NeuralUCB, which they have attributed to the fact that NeuralUCB0 does not use NNs for reward prediction.
These two concerns here have broader implications, regarding whether it's useful/necessary to explicitly use an NN to predict the reward.
- The technical contribution of the paper may be limited, since the proposed algorithm is a straightforward combination of existing methods.
- Section 3.2: If I understand correctly, for the disjoint model, when estimating the posterior distribution for the reward of an arm, you only use the previous observations collected for this arm? So the observations from all other arms are not used? Isn't this a waste of data? Please clarify whether I misunderstood.
- It's unclear to me which step of Algorithm 1 requires training a neural network?
- (minor) Section 2, first paragraph, third line: "inite-width" should be "infinite-width".
- (minor) Section 2.1, second last paragraph, first line: UCB is in fact not a "stochastic" policy.

Other comments:
- A concurrent work (paper [a] below) has also empirically evaluated neural kernel bandit algorithms in other real-world problems of autoML and reinforcement learning, and hence should be referenced.
[a] Sample-Then-Optimize batch neural Thompson sampling, NeurIPS 2022.

**Summary Of The Paper:**

This paper performs an empirical study of neural kernel bandit algorithms, to investigate the efficacy of these algorithms from different aspects such as representation learning and exploration ability. To achieve this, the paper has proposed a novel benchmark for contextual bandits which allows separately evaluating the efficacy of an algorithm in terms of representation learning and exploration.

**Summary Of The Review:**

I think the paper provides some important empirical insights regarding neural kernel bandits. I have an important concern which is the first one listed under "Weaknesses" above, which is regarding some discrepancies with the observations from recent works on neural bandits which explicitly use NNs for reward prediction. If this concern is addressed well, I'll be happy to increase my evaluation.

---

> ### Author Response · Authors · 2022-11-17
> **Clarification regarding NeuralUCB discrepancy**
>
> Thank you for the insightful comments. We appreciate the time and effort you put in to provide us with this valuable feedback. At this point, we are seeking clarification on one of our biggest points of concern, namely the relation of our algorithm to $NeuralUCB_0$. Similar points were also raised by other reviewers.
>
> **Comment: Discrepancy in performance between NeuralUCB and NK bandits.**
>
> Reported scores for NeuralUCB and NeuralTS follow directly the ones reported in (Nabati 2021 and Zhou and Zhang 2020). In addition, [(Nabati 2021)](https://arxiv.org/abs/2102.03799) has found a similar discrepancy (Table 1) between algorithms reported by [(Riqueleme 2018)](https://arxiv.org/abs/1802.09127)  and (Zhou and Zhang 2020) for the UCI datasets. We believe those discrepancies could be caused by (1) random feature approximation of NTK, in particular the number of used NN nodes (100 nodes per layer in the UCI experiment), (2) diagonal approximation of the covariance matrix, and (3) the usage of conjugate priors in linear and neural models by (Riquelme 2018 and Nabati 2021).
>
> **Comment: NNs are important for achieving good performance in neural bandits.**
>
> We think that neither NNs nor neural kernels are ultimately better than other, in the context of bandits, but rather that they have different strengths that result in better or worse performance, depending on the application. In fact (Zhou and Zhang 2020) write in their paper that: " [NTK] variant has a comparable regret bound as NeuralUCB." While we agree that the NNs are more flexible to represent wider classes of functions, we think the practical performance can come into question in applications like bandits, where the full predictive distribution can be directly utilized. A neural kernel approach provides the exact form of a distribution representing a NN, while NN approach relies on approximations. We believe the former to be a strong advantage for NK bandits.
>
> Neural kernels have also been shown to provide competitive performance in point estimate problems (e.g. [(Lee 2020)](https://arxiv.org/abs/2007.15801)). While NKs will not always perform better than their NN counterparts, it has been shown that this can indeed be a common result in applications requiring simple fully-connected architectures [(Lee 2020)](https://arxiv.org/abs/2007.15801). Our contribution is to expose the characteristics of sequential decision making datasets, for which neural kernels perform better. We also argue that utilizing the kernels directly can improve performance over the approximations proposed in $NeuralUCB_0$ (mentioned in the previous comment).
>
> **Other comments**
>
> We find the reviewer's other points clear, and we are planning to address them in a future revision of the paper.

---

### Official Review · Reviewer_tHKv · 2022-10-25

**Confidence:** 3
**Correctness:** 3
**Technical Novelty And Significance:** 2
**Empirical Novelty And Significance:** 2
**Recommendation:** 3

**Clarity, Quality, Novelty And Reproducibility:**

I believe the clarity of the paper can be improved, by formally introducing and defining the neural kernel bandit algorithm more clearly and earlier in the paper. Otherwise it is largely free of typos, and reads OK.

In my opinion, the results are not significantly novel, although there is some value in carrying out an empirical study of NK bandits. However, I am not convinced that these results extend to larger scale datasets and problem settings, and I worry about the scaling issues in the considered algorithm. Both of these are not adequately addressed in the current paper, and I believe are the hard problems to study in practice.

I have not checked the reproducibility of the experiments.

**Strength And Weaknesses:**

The paper consider NK bandits, or as studied in [1], termed as NeuralUCB_0 type algorithms.
In the highly cited paper [1], it is shown that when the neural network representation is allowed to change while training, and the changing representation is used to compute uncertainty estimates, the algorithm outperforms the fixed kernel representation based algorithms. This is in contrast to the results presented in this paper, which show the opposite phenomenon.

As the scale of the problem grows, the authors themselves remark in the paper that the complexity of computing the neural kernel is significant. However, the proposed solutions in the paper to remedy this issue are not satisfactory in my opinion.

I also generally find it uncomfortable that the authors study an algorithm which does not use the ability of a neural network to learn a good representation for a problem, and rather use a fixed representation that an infinitely wide network would realize. While there are theorems showing equivalence in a highly overparameterized limit, this is not the typical limit in which practical multi-layer neural network architectures operate. As such the studied algorithm is inherently linear, and so I do not expect it to outperform nonlinear bandit algorithms as the problem scale grows.


[1]: https://arxiv.org/pdf/1911.04462.pdf

**Summary Of The Paper:**

In this paper the authors propose an empirical analysis of neural kernel bandits, proposed as a UCB based alternative to Bayesian neural networks. Bayesian NNs have a high computational requirement as often an ensemble must be maintained in order to bootstrap a sample from the posterior, as in the case of Thompson sampling. The paper studies a version of kernel UCB where the kernel is realized by a linearized neural network in the limit where the width of the network goes to $\infty$. The experiments compare against other neural network based bandit algorithms and show that there is a benefit for using NK bandits in terms of generalization ability and computational requirements. th authors also propose a practical setting where the ability of the algorithm to learn representations is studied and show a benefit for using NK bandits here.

**Summary Of The Review:**

- I believe that the studied algorithm is not super significant and misses the point of neural network function approximation.
- I believe that there are scaling issues with the algorithm that cannot be easily resolved in spite of some proposed approaches. I think it requires a larger scale empirical study to assess whether these algorithms actually are statistically and computationally efficient.

Given these two facts, I propose a reject/weak-reject rating for the paper.

---

> ### Author Response · Authors · 2022-11-17
> **Clarification regarding NeuralUCB0 discrepancy**
>
> First, we would like to thank the reviewer for their thoughtful response. We appreciate the time and effort they put into making detailed comments. At this point, we are looking for clarification regarding the apparent discrepancies between the conclusions of our work and (Zhou and Zhang 2020). Similar points were also raised by other reviewers.
>
> **Comment: Neural networks were shown to outperform fixed kernel representations in bandits (Zhou and Zhang 2020), while we show the opposite phenomenon.**
>
> We believe there are a few reasons for the mentioned discrepancy:
>
> (1) To the best of our knowledge, no explicit use of NTK nor NNGP was recorded in the experiments by (Zhou and Zhang 2020). Instead, in the $NeuralUCB_0$ algorithm, random features were used to approximate the kernel in the primary (feature) space. While this approach is equivalent to NTKGP in the infinite network width limit, it introduces an additional discrepancy that may lower performance in practice.
>
> Moreover, in the empirical evaluation by (Zhou and Zhang 2020) the covariance matrix was diagonalized, which we believe significantly affected the performance of both $NeuralUCB$ and $NeuralUCB_0$. Diagonalization was used to save memory and computational time of storing and inverting the num\_param X num\_param matrix due to a large number of parameters. As we use the actual kernels, we don't face the same problem, given, of course, that the dataset is small.
>
> (2) Even though Zhou and Zhang (2020) provided an empirical evaluation showing the superiority of NN models over neural kernels, their theoretical analysis shows that the two approaches are comparable in terms of the bound on regret.
>
> (3) In our work we argue that utilizing the closed forms of predictive distributions does indeed improve performance over the approximations proposed in NeuralUCB/TS. A direct correspondence has been established between random infinite-width NNs and neural kernels. This correspondence enables exact Bayesian inference for regression using neural networks. That is, the samples drawn from the predictive distribution correspond directly to those drawn from a random neural network. As far as we are aware, such correspondence was not established for finite-width networks. There is promising work on approximating neural kernels with random features (e.g. Zandieh 2021), but in our preliminary tests, we have seen them perform consistently worse than their exact counterparts. We have left the comparison with random feature kernels as part of the future work, but we acknowledge the usefulness of such comparison in shedding the light on the apparent discrepancy in results between $NeuralUCB_0$ and our approach.
>
> To further address the concerns, we propose to enhance our experiments by adding our own implementation of $NeuralUCB_0$ that follows the same setup as other algorithms in our framework.
>
> **Other comments**
>
> We find the reviewer's other points clear, and we are planning to address them in a future revision of the paper.

---

### Official Review · Reviewer_st6H · 2022-10-27

**Confidence:** 3
**Correctness:** 3
**Technical Novelty And Significance:** 3
**Empirical Novelty And Significance:** 2
**Recommendation:** 5

**Clarity, Quality, Novelty And Reproducibility:**

Evaluation/comments on novelty is referred to **strength/weakness**.

**Clarity, Quality**: Though the rest of this paper is written clearly, the sections related to the experiment design (3 and 4) are somewhat difficult to follow for ones who are not familiar with the couple of papers cited in line: better to pull up some technical/mathematical details from the appendix.

**Reproducibility**: Currently the code link is hidden, but I’d like to trust the authors that their results are reproducible.,


**Strength And Weaknesses:**

**Strengths**:
1. It gives a comprehensive side by side comparison between different bandit policies on nonlinear structure data: methods include NK bandits (multiple variants),  neural-linear (LiM2) bandits, neural bandits, linear TS / UCB and multitask GP.
2. Proposes a way to measure one algorithm’s ability to learn representation and the ability to do exploration, by regulating the bandit environment with the tuple $(\epsilon, \delta)$, whose two components are controlling the learning complexity and the exploration urgency respectively.

**Weaknesses**:
1. Algorithmic novelty: it feels to me that the NK bandit policies proposed in this work are not substantially different from existing NK bandits, especially the $NeuralUCB_0$ by [Zhou et al.](http://arxiv.org/abs/1911.04462) in Appendix E, except for using separate kernels for different actions. There is some discussion in section 3.1 addressing this difference, but not sure if the separate kernel is a substantially improved design for general non-linear bandits or only fits for classification-converted bandits.
2. Results interpretation: While the paper gives a thorough description of experiment results, the interpretation is less inconclusive. Also, the wheel dataset designed to separate representation learning and exploration is somehow simple in structure due to the low-dimensionality in context ( which I assume to be $2$ if directly following the setup in [Riquelme et al.](http://arxiv.org/abs/1802.09127)). So it’s questionable whether the results give insights of NK bandits that generally apply to applications.


**Summary Of The Paper:**

Established in deep learning literature that Neural Network (NN) is related to Neural Kernels (NK), especially between the optimization dynamic of infinitely wide NN has shown to be mostly captured by the NTK at initialization. Motivated by the correspondence, this paper proposed to guide bandit policy with NK-GP (or NK-TP to take account for varying noise level in feedback), which gives both reward predictions and uncertainty estimations at the same time. Their main algorithm and its variants are referred to as NK bandits.

Empirically, this work shows NK bandits outperform other baselines on most of the tested UCI datasets.

As another claimed contribution, this paper designed a framework to empirically measure one bandit algorithm’s ability to learn representation and that to do exploration separately.



**Summary Of The Review:**

This work makes contribution by empirically comparing different bandit policies on nonlinear data and proposing a novel way to indepedently measure representation learn and exploration as well. But it is limited by not providing substantially improved method from existing work for the purpose of solving nonlinear bandits and not much insights on NK bandits that generally apply to applications were drawn from the experimental results.

---

### Decision · Program_Chairs · 2023-01-20

**Decision:**

Reject

**Justification For Why Not Higher Score:**

Because of the lack of novelty and the insufficiency of experimental comparisons.

**Justification For Why Not Lower Score:**

Can't go lower.

**Metareview: Summary, Strengths And Weaknesses:**

The reviewers commented that this paper lacks novelty vis-a-vis NeuralUCB and NeuralTS. They also mentioned that this paper boils down to a non-extensive experimental comparison among various kernels. All recommended either a reject or weak reject. The authors' rebuttals did not manage to sway the opinion of the reviewers.

**Summary Of Ac-Reviewer Meeting:**

NIL